# How does the downstream boundary affect avulsion dynamics in a laboratory bifurcation?

Gerard Salter [1], Vaughan R. Voller [2], and Chris Paola [1]

[1]Department of Earth Sciences, University of Minnesota
[2]Department of Civil, Enviornmental, and Geo-Engineering, University of Minnesota

**Correspondence:** Gerard Salter (salte040@umn.edu)

**Abstract.** Bifurcations play a major role in the evolution of landscapes by controlling how fluxes such as water and sediment are partitioned in distributary and multi-thread channel networks. In this paper, we present the first experimental investigation on the effect of the downstream boundary on bifurcations. Our experiments in a fixed-wall Y-shaped flume consist of three phases: progradation, transitional, and bypass; the first two phases are net depositional, whereas during the third, the sediment flux exiting the downstream boundary matches the input on average. We find that deposition qualitatively changes bifurcation dynamics; we observe frequent switching in the discharge partitioning under net depositional conditions, whereas bypass results in long periods of time where one branch captures most of the flow. We compare our results with a previously developed model for the effect of deposition on bifurcation dynamics. The switching dynamics we observe are more irregular and complex than those predicted by the model. Furthermore, while we observe long periods of time where one branch dominates under bypass conditions, these are not permanent, unlike in the model. We propose that the range of switching timescales we observe arises from a complex interplay of downstream-controlled avulsion, and the effect of bars in the upstream-channel, including previously unrecognized long-timescale dynamics associated with a steady bar. Finally, we describe bifurcation experiments conducted with sand but no water. These experiments share the essential feedbacks of our fluvial bifurcation experiments, but do not include bars. In these experiments, we find that the sandpile grows symmetrically while it progrades, but bypass leads to one branch permanently capturing all avalanches. We conclude that the downstream control of deposition vs. bypass is likely a major influence on bifurcation dynamics across a range of physical systems, from river deltas to talus slopes.

## 1 Introduction

Fluvial bifurcations are common in depositional systems such as deltas and alluvial fans, as well as braided and anastamosing rivers (Kleinhans et al., 2013). Non-fluvial bifurcations are also observed on debris flow fans (De Haas et al., 2018), submarine channels (Limaye et al., 2018), and lava flow networks (Dietterich and Cashman, 2014). Bifurcations are the 'switches' that steer fluxes such as water and sediment through distributary and multi-thread channel networks, controlling the evolution and form of the resulting landscape — for instance, controlling where land is built and/or maintained in river deltas.

Although bifurcations are common in depositional systems, most previous studies on bifurcation flow partitioning (Bolla Pittaluga et al., 2003; Edmonds and Slingerland, 2008; Bolla Pittaluga et al., 2015; Redolfi et al., 2016) have assumed no net

deposition (i.e. sediment bypass). This work has shown that bifurcations are susceptible to internal instability, where given a small initial perturbation, the bed morphology of the bifurcation evolves in such a way that an initially symmetric discharge partitioning becomes increasingly asymmetric. Depending on the interplay between a positive feedback favoring the deepening of one branch, and the negative feedback of cross-stream sediment transport, a symmetric bifurcation is either stable or unstable; in the latter case there may be a stable asymmetric configuration. Here, we use "stability" in the mathematical sense: an equilibrium is unstable if a small perturbation causes the system to diverge from that equilibrium (for example, a pencil balancing on its point is in unstable equilibrium), and stable if the system returns to its original state after a small perturbation (e.g., the pencil resting on its side). The stability is affected by multiple parameters, but arguably the most important is the width-to-depth ratio (i.e. aspect ratio). At low values of the width-to-depth ratio, symmetric bifurcations are stable; at high values, they become unstable.

While this aforementioned work assumes a symmetric bifurcation geometry and forcing, additional work has shown that asymmetry in boundary conditions produces a bias in the flux distribution. For example, Kleinhans et al. (2008) showed that an upstream meander steers flow in a bifurcation, and Bertoldi et al. (2009) studied the role of migrating bars in steering the flow. Imposing a difference in the downstream slopes also results in a bias (Bolla Pittaluga et al., 2003). Redolfi et al. (2019) synthesized the contribution of internal instability versus external forcing, showing that bifurcations without internal instability are completely controlled by the forcing, but when internal instability is present, for small forcings, bistability occurs, whereas for large forcing, only a single solution exists.

Edmonds et al. (2011) observed that deltas distribute sediment flux asymmetrically between their branches, which would be expected to lead to a highly asymmetric shoreline as branches receiving more sediment prograded faster. To prevent runaway asymmetry in the shoreline, they hypothesized that over time, a delta might redistribute its flux without creating or destroying any branches, a process they termed "soft avulsion."

To explore this effect, Salter et al. (2018) (hereafter SPV) added deposition to the theoretical model of Bolla Pittaluga et al. (2003). Bolla Pittaluga et al. (2003) showed that without net deposition, a bifurcation evolves towards a frozen state, either a stable symmetric state or a stable asymmetric state arising when the symmetric state becomes unstable. However, SPV showed that system-wide deposition produces ongoing avulsion dynamics, with behavior ranging from stable symmetry to soft avulsion up to full avulsion, depending on the parameters of the system. Deposition has a similar qualitative effect regardless of whether it occurs due to progradation or via purely vertical aggradation. Under symmetric forcing conditions, deposition prevents the occurrence of a steady asymmetric state.

Bertoldi and Tubino (2007) ran a set of bifurcation experiments in a Y-shaped fixed-wall flume with an erodible bed. They ran several experiments across a range of upstream channel aspect ratios and Shields stresses. They found that experiments with relatively high Shields stresses and/or low aspect ratios maintained a close to symmetric discharge partitioning. On the other hand, experiments with a larger aspect ratio and/or lower Shields stress became asymmetric. In some of these experiments, the bifurcation evolved towards a stable asymmetric state, where the discharge partitioning remained close to constant indefinitely. However, other experiments displayed large oscillations in the discharge partitioning associated with migrating bars. In these

experiments, the branch capturing the majority of the discharge switched back and forth during the course of an individual experiment, alternating between asymmetric states in which one branch was favored or the other.

Bertoldi et al. (2009) performed additional experiments and theoretical analysis to better understand the role of migrating bars in the dynamics of bifurcation flow partitioning. On the basis of their theory, and with strong agreement from their experiments, they classified bifurcations under the influence of migrating bars into four categories: "balanced," where the bifurcation remains close to symmetric and migrating bars induce only small oscillations in the discharge partitioning; "bar-perturbed," where bars induce small oscillations about an asymmetric state; "bar-dominated," where bars induce large oscillations in the flow partitioning, causing the asymmetry to switch back and forth from one branch to the other; and "one branch closed", where all of the discharge goes down a single branch. They found that the boundaries of these four phase regions were set primarily by the aspect ratio and Shields stress of the upstream branch.

In this paper, we investigate the role of the downstream boundary in controlling the dynamics in a set of laboratory fluvial bifurcation experiments, which for the first time include the effect of system-wide net deposition. Our experiments included both net-depositional and bypass stages, allowing us to isolate the effect of deposition on the dynamics of the bifurcation, and thereby test the theory of SPV. Consistent with the theory, we find that deposition results in more frequent avulsions, whereas bypass leads to long periods of time where one branch captured the majority of the flow. However, unlike in the SPV theory, bypass does not stop avulsion entirely. We also find that the avulsion dynamics are significantly more complex than the simple oscillations predicted by theory. Additionally, to remove the effect of bars and extend our work to non-fluvial bifurcations, we ran an experiment with sand but no water, and found that avalanches occurred in both branches during progradation, but ultimately went down a single branch under bypass. Overall, our results indicate that deposition increases the frequency of bifurcation avulsion, but rich switching dynamics occur even without net deposition.

## 2 Fluvial Bifurcation Methods

### 2.1 Experiment Design

We performed a series of five experiments in a fixed-wall Y-shaped flume (Fig. 1). The upstream branch was 0.9 m long, and the two downstream branches were 2.1 m long measured along the outside wall. Roughly following hydraulic geometry scaling, the upstream branch was 0.1 m wide, and the two downstream branches were both 0.07 m wide. The angle between the two downstream branches was $16°$, which is a relatively low angle compared to delta bifurcations in nature (Coffey and Shaw, 2017). We selected a narrow angle due to space constraints and to reduce shadows in the topographic scans. We note that angle does not play a role in the quasi-1D model of Bolla Pittaluga et al. (2003), and Thomas et al. (2011) found that angle has little effect on discharge partitioning in fixed-bed experiments. Water and sediment were introduced from upstream and fed directly into a rock crib, which diffused the flow. In each branch, a 30 mm high broad-crested weir acted as the downstream boundary condition. We leveled the table supporting the flume such that when no sediment was fed, the water discharges in each branch were nearly equal.

**Figure 1.** a) Schematic side-view of an experiment. As water and sediment are fed from upstream, the delta deposit grows and progrades until it reaches the downstream weir. The downstream weir is also responsible for setting the base level. b) Overhead image from an experiment, showing the geometry of the Y-shaped bifurcation flume.

The purpose of our experiments was to test the effect of net deposition on bifurcation dynamics. Our usage of the term "deposition" here refers to the system-wide mass balance condition that the sediment flux input exceeds the sediment flux exiting the downstream boundary. In contrast, we define "bypass" as the condition that the sediment flux input and output are in balance on average. Because the sediment flux input from upstream is fixed, the downstream boundary controls whether the system is net-depositional or in bypass. We note that local deposition (e.g. due to bar migration) can occur even when the system is in a bypass state. Each experiment consisted of three phases. Starting from an empty flume with no initial slope, we fed water and sediment continuously from upstream. In all three experiment phases, water freely runs off over the downstream weirs. In the first phase of the experiment, progradation, all sediment is trapped within a growing delta that progrades down each branch, and therefore no sediment exits over the downstream weirs. In the second phase of the experiment, the transitional stage, the delta fronts reach the downstream weirs and some sediment begins to exit over the downstream weirs, but the sediment output is smaller than the sediment input, meaning that the system is still net-depositional. Finally, the last phase of the experiment is bypass, where on average all of the sediment fed from upstream exits downstream.

We used 0.3 mm quartz sand in all of our experiments. Table 1 shows the run parameters for each of the five experiments. The only variables we changed between experiments were the input water and sediment fluxes. The first four experiments have approximately the same sediment flux but different water discharges; the fifth has a lower sediment flux. We also include in Table 1 several measured dependent variables, which represent spatially and temporally averaged values over the upstream portion of the flume, measured once the system had reached bypass.

| Experiment | $Q_w$ (mL/s) | $Q_s$ (mL/s) | $h$ (mm) | $S$ (-) | $W/h$ (-) | $q_{s*}$ (-) |
|------------|--------------|--------------|----------|---------|-----------|--------------|
| exp 1 | 182 | 0.61 | 4.7 | 0.027 | 21 | 0.29 |
| exp 2 | 140 | 0.61 | 3.6 | 0.033 | 28 | 0.29 |
| exp 3 | 103 | 0.60 | 2.3 | 0.054 | 44 | 0.29 |
| exp 4 | 46 | 0.52 | 1.4 | 0.071 | 72 | 0.25 |
| exp 5 | 125 | 0.29 | 2.9 | 0.042 | 35 | 0.14 |

**Table 1.** Parameter values for each run. The independent variables are water discharge $Q_w$ and volumetric sediment flux $Q_s$. Dependent variables are depth $h$ and slope $S$, measured over the upstream 20 cm of scans once the system reached bypass. Aspect ratio $W/h$ and dimensionless sediment flux $q_{s*}$ are also listed, where the dimensionless sediment flux is defined $q_{s*} = \frac{Q_s}{W} \left( d_s^3 g \frac{\rho_s - \rho}{\rho} \right)^{-1/2}$, $d_s$ is the grain size, $g$ is gravitational acceleration, $\rho_s$ is the sediment material density, and $\rho$ is the density of water.

## 2.2 Data Acquisition

We took time lapse images from overhead and orthorectified them using a grid of control points and Matlab's projective image transformation. We dyed the water blue so that the qualitative depth pattern was visible; in our images deeper areas are a deeper blue. To enhance the contrast between deep and shallow flow, we compute the color index $C$:

$$C = \frac{B - R}{B + R} \tag{1}$$

where $B$ and $R$ are the intensities of the blue and red bands of the image, respectively. Examples of $C$ obtained from overhead images are shown in Fig. 2.

Using the images, we obtained a time series of the sediment front position (Figure S1 of the supplemental info). This was done automatically by thresholding a differenced image between the current time and a reference time where no deposit was present, isolating pixels within the delta deposit. We then extracted the boundary of the binarized delta deposit, and identified the pixels corresponding to the sediment front. We averaged the positions of the sediment front pixels in the center 2-3 cm of each branch, and projected the position onto a line that was parallel to each branch, with the origin at the branch entrance. We defined the start of the progradation phase as the time at which the sediment fronts reached the bifurcation point, and the start of the transitional phase as the time at which the first sediment front reached the downstream weir.

We used a laser sheet system to measure the topography of the experiment with 1 mm horizontal resolution and sub-millimeter vertical accuracy. We turned off the experiment periodically to measure the bed surface topography. Immediately before turning off the experiment, we also measured the water surface elevation. This was done by introducing a slurry of water

and titanium dioxide from upstream, which turned the water opaque, allowing the scanner to measure the water surface. We obtained the depth by subtracting the bed scan from the water surface scan. Fig. 2 shows a comparison between the qualitative pattern obtained by computing $C$ from overhead images and the quantitative measure of depth obtained from scans. The approximate time interval between scans was 2 hours, whereas the time interval between images is 30 seconds.

The sediment discharges at the downstream end of both branches were measured continuously. Below each overfall, $\sim 19$ liter buckets received the water and sediment exiting the experiment. Each bucket was placed on a logging scale. Water was free to flow out over the top of the bucket, but the sediment settled out within the bucket. Because the density of the sediment $\rho_s$ was 2.65 times the density of water $\rho$, and the total volume of water and sediment was roughly constant, we use the mass $M$ on the scale to calculate the mass of sediment $M_s$ as:

$$M_s = \frac{\rho_s}{\rho_s - \rho} (M - M_0) \tag{2}$$

where $M_0$ is the mass if the entire volume on the scale were water. We applied a small correction to $M_0$ to account for slight variation in water level as a function of discharge using the parts of our time series where the sediment flux was known to be zero. We smoothed the time series of sediment mass with a moving average filter, and took the time derivative to obtain the sediment flux.

We also measured water discharge continuously. In each branch, water exiting the sediment collection bucket entered a cylinder with a small horizontal hole near the bottom. An outer cylinder prevented splash, and the water was allowed to exit from the outer cylinder to the drain through a relatively large opening to prevent water from pooling in the outer cylinder. The entire assembly was mounted on a logging scale. The flow rate out of the hole in the inner cylinder is a function of the water level in the inner cylinder. Using a calibration curve, we computed water discharge using the mass on the scale.

Given that the incoming discharge was constant, one of the discharge measurement devices is redundant, as the discharges of the two branches should sum to a constant value. We observed some deviations from constancy, associated with error in the calibration curve, and with one of the measurement devices, which did not lock on perfectly to the scale causing it to shift slightly over time. However, these errors are small enough that they do not affect the interpretation of our results. Additionally, in one experiment the maximum measurable discharge was triggered at times, in which case we had to rely on the measurement
from a single scale.

## 3   Prediction Based on Salter et al. (2018) Model

The model of Bolla Pittaluga et al. (2003) (hereafter BRT), which assumes bypass conditions, predicts that a bifurcation will evolve towards an unchanging discharge partitioning. This partitioning could be either symmetric or asymmetric, with a tendency towards greater asymmetry under a higher upstream channel aspect ratio or lower Shields stress. SPV added the effect
of deposition to the BRT model. This resulted in a range of dynamics, from stable symmetry to soft avulsion and full avulsion, depending on parameters of the system. A key prediction from the model is that depositional bifurcations undergo frequent switching, whereas bypass leads to an unchanging discharge partitioning. However, we note that SPV did not include the effect

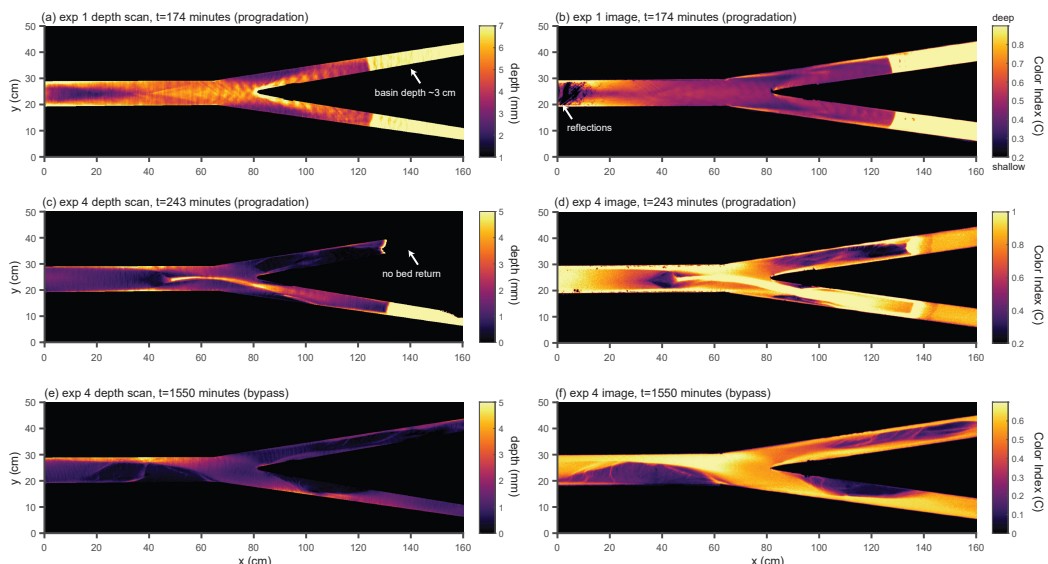

**Figure 2.** Comparison between water depth obtained from topographic scans (a, c, and e), and the Color Index $C$ obtained from overhead images (b, d, f). Although the $C$ does not map linearly to the water depth, it qualitatively reproduces the flow depth patterns. Examples are taken from the progradation phase of experiment 1 (a, b), progradation phase of experiment 4 (c, d), and bypass phase of experiment 4 (e, f).

of migrating bars in the model. Bertoldi et al. (2009) coupled the effect of bar migration to the BRT model, and found that this can allow for oscillations under bypass conditions.

For reference, we run the SPV model using the same geometry and run parameters as our experiments. A full description of the model is found in Salter et al. (2018), but in short, we model the bifurcation branches in one dimension, and divide the reach immediately upstream of the bifurcation into two laterally interacting cells. The length in the longitudinal direction of these two cells is an important parameter of the model, and is given by $\alpha W$, where $W$ is the upstream channel width, and $\alpha$ is an order-one parameter. Transverse exchange of sediment between these two cells occurs as a result of cross-stream flow and the presence of a transverse bed slope; the strength of the latter control is specified through the parameter $r$. The whole system, including the two laterally interacting cells, upstream feeder channel, and two downstream branches, evolves using sediment transport and mass conservation equations. Variation in the lateral bed slope upstream of the bifurcation and longitudinal bed slopes into the branches allows for discharge partitioning dynamics.

Like our experiments, our numerical runs consist of three phases: prograding, transitional, and bypass. We start from a symmetric initial condition with a small initial perturbation. During the first phase, once switching of the discharge partitioning starts, it occurs regularly, with a gradual increase in amplitude and period. Then, starting in the transitional period, the dynamics slow dramatically, and under bypass conditions dynamics stop entirely. The final configuration under bypass conditions is a frozen state, with a discharge asymmetry that depends primarily on the aspect ratio and Shields stress of the upstream channel. We note that in the model, once a sediment transport formula is chosen, choosing a value for the Shields stress is equivalent

to setting the dimensionless sediment flux. When comparing the SPV model results with experiments, we prefer to use the dimensionless sediment flux instead of the Shields stress, because it is the independent variable, and our estimation of the Shields stress is complicated by non-uniformity in the upstream channel (see supplemental info). In the model, we observe greater asymmetry, all else equal, for larger aspect ratios and lower dimensionless sediment flux, and greater asymmetry with bypass compared to deposition. Figure 3 shows an example of a model run.

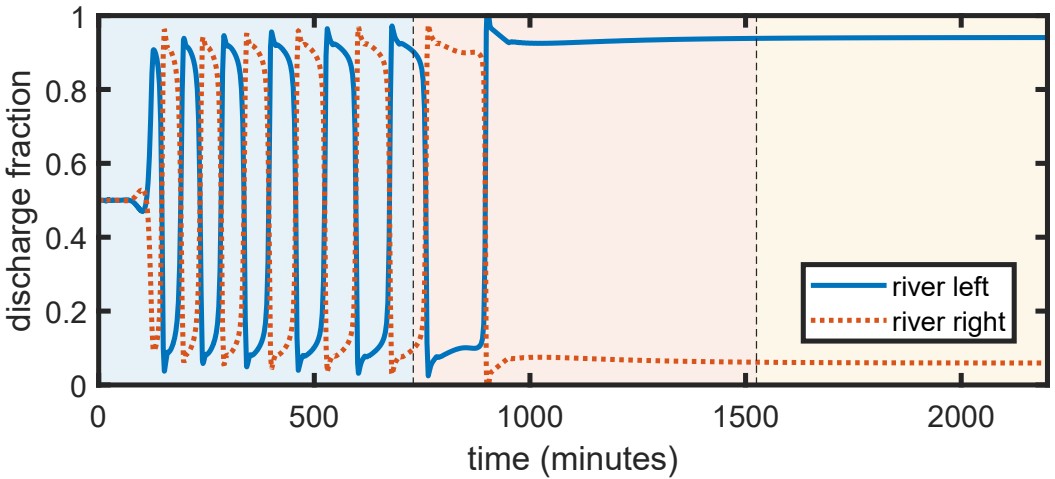

**Figure 3.** Typical water discharge partitioning time series obtained from the SPV model. Background colors represent the progradational, transitional, and bypass portions of the experiment, respectively.

In the next section, we describe the results from our experiments, and compare the results to the predictions of the SPV model.

## 4   Fluvial Bifurcation Experiments

As stated previously, each experiment consists of three phases: progradation, transition, and bypass. After an initial building period that is not relevant to our study, the start of the first phase occurs when the delta deposit first enters the 'Y' portion of the flume, as determined from overhead images. Once one of the two branches reaches the downstream weir, the next phase begins. The length of this transitional phase varies between experiments. Due to variability in the total output sediment flux, and the fact that the approach to bypass is gradual, determining the change from transitional to bypass phases is not obvious. We defined the start of the bypass stage as the point where the output sediment flux is within one standard deviation of the average sediment flux of the bypass stage. The sediment flux time series for each experiment are shown in Fig. 4, and the water discharge asymmetry time series are shown in Fig. 5.

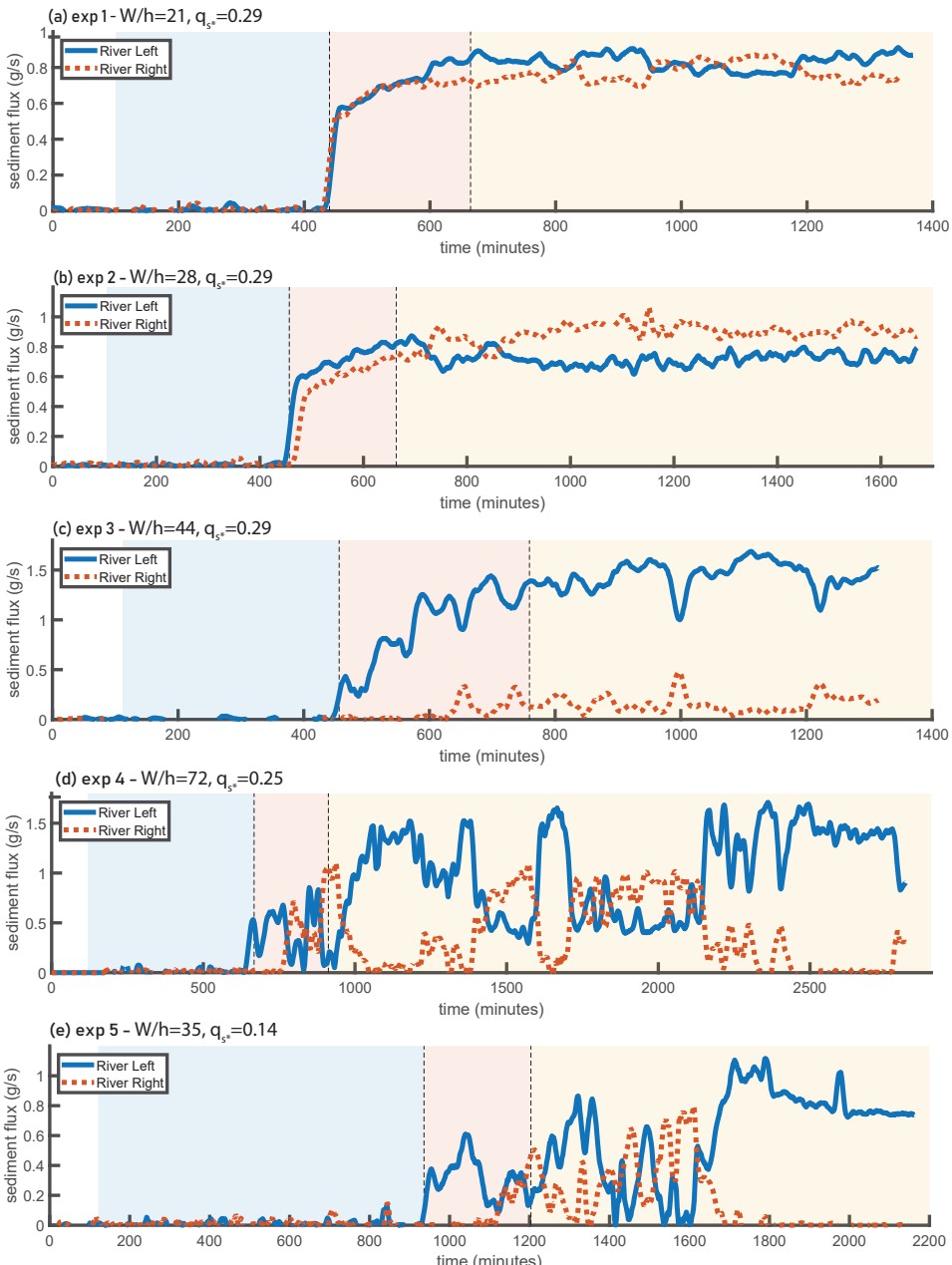

**Figure 4.** Time series of sediment flux exiting each branch. The data have been smoothed using a double moving average with a window of 20 minutes. Background colors represent the progradational, transitional, and bypass phases of each experiment. During the progradation portion of the experiments, no sediment leaves the flume. The outgoing sediment flux gradually increases during the transitional phase, and reaches a statistical steady state during the bypass phase.

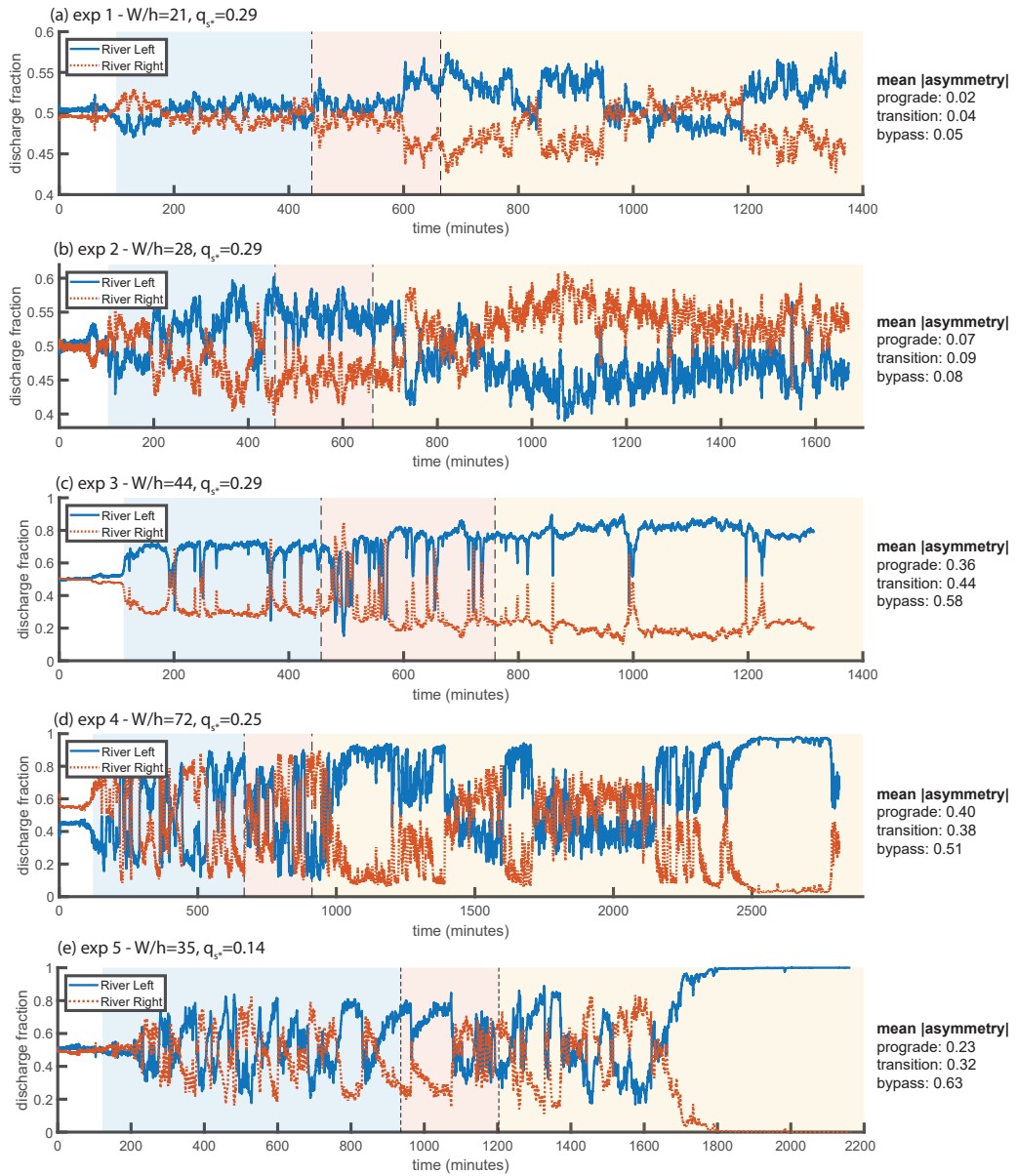

**Figure 5.** Time series of the water discharge fraction exiting each branch. Background colors represent the progradational, transitional, and bypass phases of each experiment. Mean values of the absolute value of the asymmetry ($|\Delta Q|$) of each phase are provided to the right of each time series. Note that $|\Delta Q|$ is equivalent to the difference between the larger discharge fraction and smaller discharge fraction.

## 4.1 Mean Asymmetry

Based on the original BRT model and our own model simulations, we predict that increasing the upstream width-to-depth ratio should increase the asymmetry of the experiment, and decreasing the dimensionless sediment flux also increases asymmetry. Following the definition from Bertoldi et al. (2009), we define the discharge asymmetry $\Delta Q$ as $\frac{Q_{RL}-Q_{RR}}{Q_{RL}+Q_{RR}}$, where $Q_{RL}$ and $Q_{RR}$ are the water discharges through the river left and river right downstream branches, respectively. In our experiments, we control the width-to-depth ratio by changing the input water discharge. As can be seen in Fig. 5, comparing experiments 1 through 4, run with approximately the same sediment flux but differing water discharges, we find that as input water discharge increases (i.e. aspect ratio decreases), asymmetry decreases. This is consistent with the model prediction. Experiment 5 was run with a lower dimensionless sediment flux. This results in greater overall asymmetry (for the given aspect ratio), consistent with the model.

## 4.2 Qualitative Description of Experiments

Experiments 1 and 2 were the closest to symmetric of the experiments. As shown in Fig. 5, some switching in the water discharge asymmetry occurs during the progradational and transitional portions of these runs, with a few switches in sign, and superimposed high frequency variability. In the bypass portion of experiment 1, there are times of measurably higher discharge asymmetry where the river left branch is favored, but the asymmetry is still small ($\Delta Q \sim 0.1$). However, these periods are interspersed with periods of milder asymmetry. The bypass portion of experiment 2 does not have these periods of relatively high asymmetry, but does have a tendency to favor the right branch. Interestingly, this was the only experiment that favored the right branch; to the extent that other experiments favored a branch, it was always the left side.

Experiment 3 was unusual in that there was a clear bias towards favoring the river left side in not just the bypass stage but for the entire duration of the experiment. Despite this preference, we do observe a change in dynamics between the progradational and bypass portions of the experiment. The progradational and transitional portions of the experiment have occasional short-duration periods where the right branch becomes highly favored; these are almost entirely absent from the bypass portion of the experiment. We also observe that the maximum asymmetry during the bypass portion of the experiment is slightly larger than during the progradational phase.

Experiment 4 had frequent high-magnitude switches during the progradational and transitional phases of the experiment. In contrast, the bypass phase contains long periods of time where the left branch captures most of the discharge (i.e in Fig. 5 approximately minutes $1000-1400$, $1600-1700$, and $2200$ onward). However, these periods are interspersed with times where the discharge asymmetry is milder. The asymmetry magnitude during these periods of milder asymmetry is actually lower than the asymmetry of the switches observed in the depostional portions of the experiment. We note that throughout the duration of the bypass portion of the experiment, there was a persistent steady bar immediately upstream of the river right branch. The dynamics we observe during the bypass portion of the experiment are associated with the partial erosion and re-formation of this bar, and are described in greater detail in the next section.

Experiment 5 was similar to experiment 4 during the progradational and transitional portions of the experiment, displaying frequent switches. Towards the end of the progradational phase and start of the transitional phase, there were a few relatively long duration switches favoring the left branch. The dynamics during the start of the bypass stage do not differ qualitatively from the preceding dynamics. However, starting around minute 1700, the left branch gains discharge, and eventually captures the entirety of the flow. There is no guarantee that a switch would not have occurred had we run the experiment longer. However, we believe that the complete avulsion was most likely permanent, due to the significant difference between the bed elevation of the right branch and the local water elevation(see supplemental figures S2 and S4). Even though we observed the growth of a region of scour in the upstream portion of the right branch, the downstream bed elevation was high enough to act as a barrier to avulsion. Permanent complete avulsions were also observed in some of the experiments of Bertoldi et al. (2009). While the frozen state we observed towards the end of the experiment is consistent with the prediction of the SPV model, the switches that occurred during the first half of the bypass stage are not.

## 4.3 Bypass Dynamics

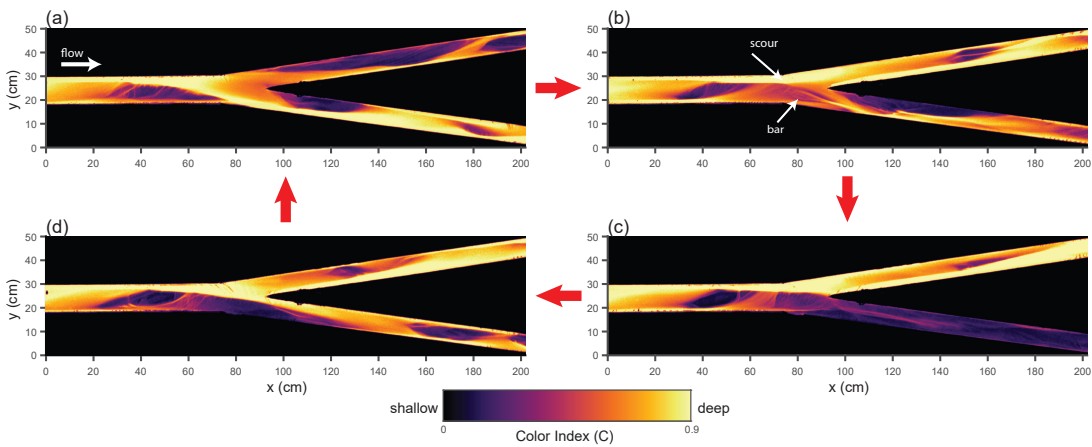

**Figure 6.** Color Index $C$ obtained from overhead images, illustrating the bypass dynamics observed in experiment 4. Images are taken at time (a) 1595 minutes (b) 1602 minutes (c) 1698 minutes (d) 1707 minutes. Corresponding values of $\Delta Q$ are: (a) $-0.55$ (b) $0.38$ (c) $0.84$ (d) $0.34$.

Next, we describe in detail the dynamics of experiment 4, which displayed a dramatic change in dynamics between the net-depositional and bypass stages, as seen in Figure 5. In this experiment, frequent switching occurs during the progradational and transitional stages of the experiment. On the other hand, during the bypass stage, the bifurcation favors the river left branch for long periods of time, appearing to become frozen. These periods coincide with the presence of a steady bar blocking the entrance to the right branch, and a region of scour at the entrance to the left branch. However, we find that the bifurcation occasionally unfreezes, and the right branch reopens and can remain open for an extended period.

These dynamics of freezing and unfreezing appear to be a previously unrecognized type of autogenic cycle, as illustrated in Fig. 6. The cycle begins with the deepening of one branch due to a positive feedback (Subfigures (a) to (b) in Fig. 6), as described in the BRT theory. The fact that in our experiment this always coincides with the river left branch is probably due to small imperfections in the construction of the bifurcation geometry. At this stage in the cycle, immediately upstream of the
bifurcation, cross-stream flow of water and sediment occurs from river right to left. Due to the nonlinearity of sediment transport, consolidation of the flow through a single branch enhances the downstream sediment flux, leading to erosion upstream of the bifurcation. This erosion causes the surface of the bar at the entrance to the right channel to become abandoned (Subfigure (c) in Fig. 6). However, cross-stream flow of sediment immediately upstream of the bifurcation appears to be important in stabilizing the scour region; as the bar surface is abandoned, the scour region widens until enough of the bar edge has been
eroded that the right branch is reopened (Subfigure (d) in Fig. 6). However, with both branches open, less sediment flux is conveyed, forcing the system to aggrade (see bed profiles in supplemental Figure S2). This allows the flow to reoccupy any remnant portions of the bar upstream of the right branch, setting up the conditions for the cycle to begin again. We observed similar dynamics associated with partial erosion and re-closure of the bar in experiments 3 and 5 as well, but the cycle was clearest in experiment 4. From Figure 5, we observe roughly 2.5 of these long cycles during the bypass portion of experiment
4 (notice long periods where the left branch dominates, followed by extended periods of reduced asymmetry), with higher frequency dynamics superimposed on the long cycles.

### 4.4 Distribution of Asymmetry Values

Figure 7 shows the distribution of the asymmetry in water discharge partitioning. For each experiment, the distributions of the two net-depositional phases are plotted separately from the distribution of the bypass phase. For all experiments except
experiment 2, we find that the distributions shift towards greater values of asymmetry under bypass conditions, with a clear preference for the left branch over the right branch. Experiments 1 and 2 have the lowest overall degree of asymmetry, but nevertheless exhibit small shifts between the distributions of the depositional and bypass phases. The distribution for experiment 3 exhibits a persistent bias towards the left side. With bypass, this bias is even stronger. For depositional conditions, experiment 4 has a broad distribution of asymmetry values. In contrast, under bypass conditions, there is a relatively narrow double peak
at very high asymmetry values favoring the left branch, and a very broad peak slightly favoring the right branch. During the bypass phase, extreme negative asymmetry values (i.e. favoring the river right branch) have diminished frequency relative to the depositional phase. Like experiment 4, the histogram of experiment 5 exhibits a shift towards high asymmetry values favoring river left between the depositional and bypass phases, but also shows a slight increase in the frequency of highly negative asymmetry favoring river right, whereas this part of the distribution was suppressed in experiment 4. The largest peak in the
distribution coincides with the complete abandonment of the right branch.

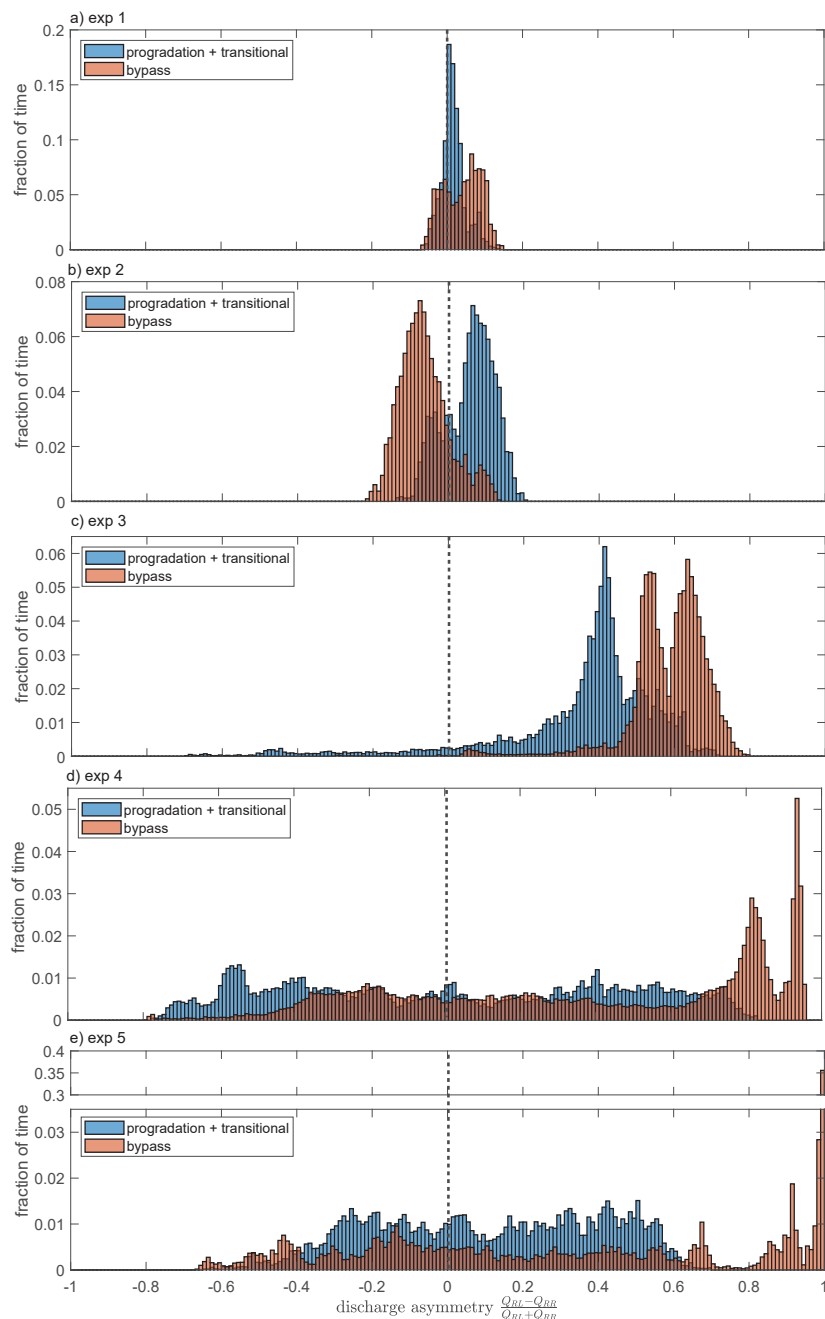

**Figure 7.** Histograms of the water discharge asymmetry from each experiment. Blue shows the progradation and transitional phases, and orange is the bypass phase. Negative values of the discharge asymmetry indicate that the river right branch is capturing a greater proportion of the discharge, and vice versa for positive values. Note break in y-axis for subfigure (e).

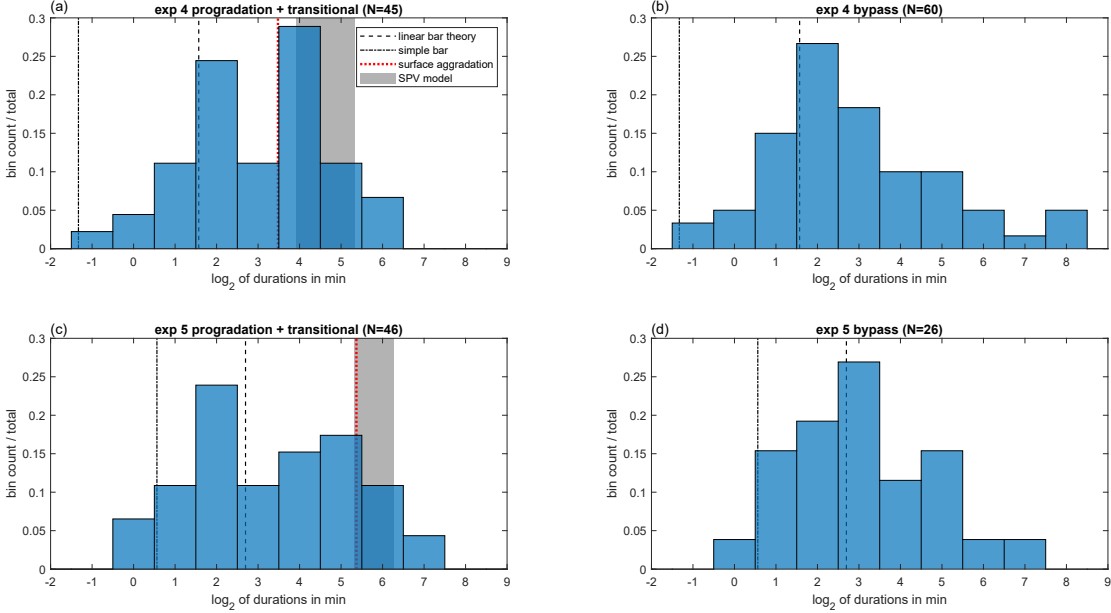

**Figure 8.** Histograms of duration between switches for (a) experiment 4 progradation and transitional phases and (b) bypass phase, and (c) experiment 5 progradation and transitional phases and (d) bypass phase. Theoretical timescales are shown for comparison.

## 4.5 Avulsion Timescales

Next, we seek to characterize the timescales of the switching behavior we observed in our experiments, and compare those timescales to those that would be predicted to arise from downstream-controlled avulsion and from migrating bar-induced switching.

5     We first used a Fast Fourier transform to look for characteristic frequencies. However, we found that there are no obvious peaks in the plot of power spectral density. The power is highest at low frequencies, and decays as approximately $1/f^2$. We also used wavelet analysis, following methods suggested by Torrence and Compo (1998), but similarly, we found that low frequencies dominated the Wavelet power spectrum, and that there were no persistent statistically significant peaks at higher frequencies.

10     Next, we looked at the statistics of switch duration. For each switch event, we start a timer when the discharge rises above an arbitrary threshold, here $55\%$ of the total discharge, and keep the timer running until the discharge falls below $45\%$ in that branch. We then record the duration, and start the timer for the next event. We repeat this procedure for the duration of our time series, obtaining a series of switch durations. While the precise distribution is sensitive to the choice of threshold, the results are qualitatively similar. We chose a threshold that is large enough that we could be confident that all switches we recorded are

15     true switches, rather than arising from noise in our measurement. We note that durations are not recorded if we do not observe the end of the event. Both experiments 4 and 5 ended after favoring one side for a long period of time; these events are not

included in the duration histograms of Fig. 8, but they would represent an additional count on the long-duration tails of the histograms.

We focused our analysis on experiments 4 and 5, because these were the experiments with the most robust and high-magnitude switching. We generated separate histograms of the switch duration for each of these two experiments with deposition (progradation and transitional phases) and bypass. The small number of switches makes drawing a meaningful distinction between the histograms impossible; using a two-sample Kolmogorov-Smirnov test or a two-sample Anderson-Darling test, we are unable to rule out the possibility that all four histograms are random realizations of the same underlying distribution. We tested the possibility that switches occur randomly according to a Poisson point process. A Poisson point process yields an exponential distribution of switch durations. If we relax the assumption of randomness and allow for short-time inhibition, the resulting distribution is a gamma, of which the exponential distribution is a special case. Using either a one-sample Anderson-Darling test or Kolmogorov-Smirnov test, we can reject the null hypothesis with $p < 0.05$ that our experimental distributions are drawn from a gamma distribution, with $p$ values calculated using a Monte Carlo simulation to account for the fact that we are estimating the parameters of the distribution from the distribution itself. On the other hand, we find that our distributions are consistent with a lognormal distribution.

With the caveat that our experiments generated a small sample size of switches, we can compare the observed switch durations with those expected from different physical mechanisms. Therefore, we seek to estimate the timescales that would be expected for downstream-controlled avulsion, and for migrating-bar induced switching.

We begin by running the SPV model to generate a prediction of the downstream-controlled avulsion timescale. We use the parameter values reported in Table 1, and the Wong and Parker (2006) correction to the Meyer-Peter and Müller (1948) sediment transport formula. We note that because the imposed dimensionless sediment flux sets the upstream Shields stress, we cannot set both the upstream depth and slope independently. We choose to match the slope rather than the depth, because the slope is a more important control on the avulsion timescale. Using the parameter values from experiment 4, we obtain a switching timescale that ranges from approximately $15$ to $40$ minutes as the branches lengthen. We note that the SPV model depends on the free parameters $\alpha$ and $r$, chosen as $\alpha = 1$ and $r = 0.5$. We find that these parameters do not strongly affect the switching timescale, but they strongly affect the degree of asymmetry and the onset time of switching. For example, when $\alpha$ is doubled, the onset of switching is delayed, but the approximate switching timescale at the end of the progradation phase is $34$ minutes. Using parameter values from experiment 5, we obtain a switching timescale that ranges from approximately $40$ to $78$ minutes. An alternative estimate for the downstream-controlled switching timescale that does not rely on the use of a numerical model is the time $T$ to fill the surface area $A$ of the experimental domain to one channel depth $h$, i.e. $T = \frac{Ah(1-\lambda)}{Q_s}$, where the porosity $\lambda$ is assumed to be $0.4$, and the surface area $A$ is roughly $0.42$ m$^2$. This estimate does not account for the volume of sediment deposited in the foreset of the prograding branches, or the change in topset area over time. Using this method, we obtain an estimated timescale of $11$ minutes for experiment 4 and $41$ minutes for experiment 5.

The second physical mechanism to assess is switching induced by downstream-migrating bars in the upstream channel, as proposed by Bertoldi et al. (2009). Our first estimate of the timescale is derived from the linear bar theory of Colombini et al. (1987). The relevant timescale is the time for the fastest-growing bar to migrate $1/2$ of the bar wavelength (the $1/2$ bar

wavelength is the distance from a bar to the next downstream bar on the opposite side of the channel). We obtain our estimate using linear bar theory code from Redolfi et al. (2019). Using parameter values from experiment 4, the angular frequency of the fastest growing bar is 0.003; this translates to a switching timescale of 3.0 minutes. Using parameter values from experiment 5, we obtain a switching timescale of 6.5 minutes. Alternatively, a simpler way to estimate the switching timescale $T$, which again is the time required for a bar to translate $1/2$ bar wavelength, is as follows: we assume a rectangular cross-section with a height equal to the channel depth $h$ and width equal to $1/2$ of the channel width $W$, a typical bar $1/2$ wavelength of $3W$ (Adami et al., 2014), and that all of the volumetric sediment flux $Q_s$ contributes to the bar (i.e. no sediment bypasses the bar). Based on these assumptions, we obtain $T = \frac{1.5W^2h(1-\lambda)}{Q_s}$. From this expression, for experiment 4, we obtain a timescale of $0.4$ minutes, and for experiment 5, we obtain a timescale of $1.5$ minutes.

We can compare these theoretical timescales with those obtained from our experiments, shown in Fig. 8. Our theoretical estimates for bar-migration timescales are on the order of minutes or less, generally falling within the shortest observed switching times in our distributions. The simple bar timescale estimates are short compared to those from linear bar theory, perhaps due to the assumption that no sediment bypasses the bar. Our estimates for the downstream-controlled avulsion timescale are in the tens of minutes, and generally fall within the longest switching times we observed. However, we note that long switch-times are observed even during the bypass portion of our experiments, where SPV theory predicts an infinite downstream-controlled avulsion timescale. The theoretical timescales for both bars and the downstream control are approximately $2-4$ times higher for experiment 5 compared to experiment 4, owing primarily to the differences in sediment flux and channel depth. Although based on statistical tests we cannot preclude that the experiment 4 and experiment 5 histograms differ purely by chance, a doubling of the switch durations (i.e a shift to the right by one bin in Figure 8) is reasonably consistent with the observed distributions.

## 5  Sand Pile Experiments

Given the complexity we observe in the experiments described above, we decided to study a system stripped to the bare essentials: a positive feedback that causes a growth in asymmetry, and the presence or absence of progradation, which due to mass balance acts as a negative feedback via the slope. To accomplish this, we ran an additional set of experiments in which sand but no water is fed from upstream, producing a sand pile. These experiments were run on a table top with straight side walls roughly 40 cm in length and 10 cm apart, and a central dividing wall starting halfway downslope (Fig. 9a). Similar to the fluvial experiments, the downslope boundary condition was a free overfall. By running without water, these experiments are free of the complexity introduced by migrating bars while preserving the essential feedback described above.

In these experiments, we started with a flat, empty initial condition and then fed sediment continuously. A growing sand pile developed, which prograded through avalanching. We found that in the initial phase of the experiment where the sand pile toe had not reached the table edge, avalanches went down both sides of the bifurcation in roughly equal proportion, with neither branch advancing far ahead of the other. Many avalanches went down both branches simultaneously. Next, the sand pile toes reached the table edge and sand began to exit over the table edge. We found that eventually, once the pile reached the end of

the table, all of the avalanches went down a single side, and the other side was completely closed off indefinitely (Fig. 9c). We also observed that the avalanches became more regular in size and frequency.

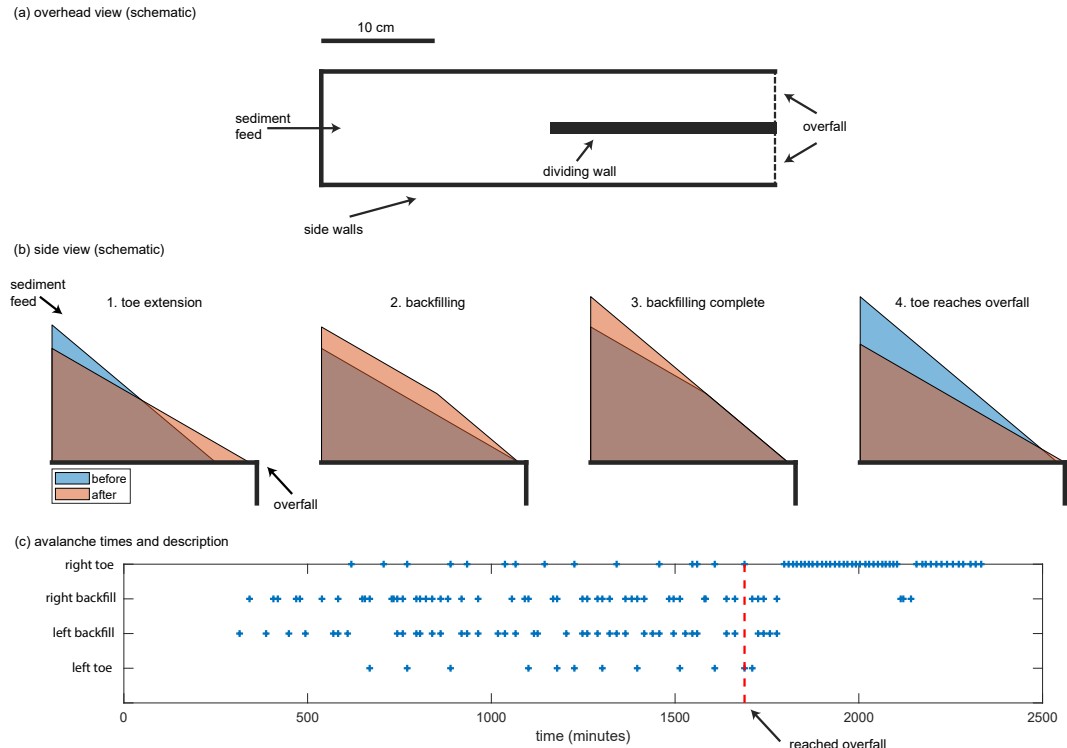

**Figure 9.** (a) Schematic of the experimental setup for the sandpile bifurcation experiments. (b) Schematic side-view illustrating stages of sandpile growth. (c) Avalanche type and location vs. time. Red dotted line is when the sandpile toe first reached the overfall (table edge).

In addition, we observed that once the system reached its final state, the bed elevation was lower upstream of the entrance to the active branch than the closed branch. This lateral slope appeared to be responsible for steering the avalanches down a single side. We hypothesize that with progradation, the lateral slope is self-correcting. We observed that when prograding, the sand pile typically grows through large avalanches that erode the top of the pile and extend the toe of the pile. In response to the extension of the toe, smaller avalanches backfill from downstream to upstream. We hypothesize that large avalanches that create a lateral slope extend the sand pile toe further, allowing subsequent backfilling avalanches to erase the lateral slope (Fig. 9b). However, once the sand pile reaches the edge of the table, large avalanches are no longer free to extend the toe of the pile. This turns off the backfilling process. Without this self-correction, a positive feedback emerges where a slight lateral slope advantage produces larger avalanches which scour deeper, creating an even larger lateral slope until all avalanches are steered down a single side.

We found that this stark difference in dynamics was not sensitive to the details of how our experiments were conducted; we observed qualitatively similar behavior at both higher and lower sediment feed rates, with different sediment types (low-density walnut shell sediment or quartz sand), and different flume geometries (we tried both narrower and wider branches). We found that the choice of side once the pile reached the table edge was sensitive to the position of the sediment feeder. When the sediment feeder was slightly off center, the avalanches inevitably followed the branch closer to the feeder. However, we found that once a side had been chosen, shifting the feeder position closer to the opposite branch did not result in a switch, even when the feeder was moved past center. Only when the feeder was shifted far past center did the avalanches switch side. This indicates that while the choice of side is highly sensitive to any asymmetric imperfections, once a side is chosen it is stable to modest perturbations.

## 6 Discussion

Our interpretation of the dynamics in our fluvial experiments is that they arise from a complex interplay between bar dynamics and the downstream control. As is clear from the bypass portion of our experiments, bars produce vigorous and complex dynamics by themselves. These include frequent oscillations induced by migrating bars, as proposed by Bertoldi et al. (2009), as well as longer-period dynamics associated with a steady bar. What then is the role played by net deposition? We propose that deposition acts as a stopgap: whenever bar dynamics alone are not sufficient to distribute sediment evenly between branches, deposition results in a self-correcting tendency via the difference in downstream slope between the two branches. In our experiments, we observed large asymmetries in the time-integrated sediment delivery to the two branches under bypass conditions. On the other hand, deposition prevents this asymmetry from growing without bound.

This view that the dynamics arise from the interplay of bars and the downstream control is supported by our sandpile experiments. In the sandpile, there was no source of internal variability to kick the system out of a frozen state once the system was in bypass. Again, deposition produced a self-correcting effect via the slope: if one branch grew a little faster than the other for any reason, the difference in downstream slopes allowed the opposite branch to catch up. Our sandpile experiments suggest that downstream-controlled avulsion is not limited to river deltas. We expect to find similar behavior in a range of physical systems, from debris flow fans to lava networks to talus slopes.

The rich bar-induced dynamics we observed during our experiments are surprising; given the relatively short length of the upstream channel ($\sim 10W$), migrating bars, if they occur, are unlikely to be fully developed (Federici and Seminara, 2003; Fujita and Muramoto, 1985). Nevertheless, at times we observed free bar migration on the timescale of minutes in the downstream portions of the upstream channels (as an example, see Fig. S3 in supplemental info). Bar migration induced oscillations in the flow, consistent with previous work (Bertoldi and Tubino, 2007; Bertoldi et al., 2009). At other times, we observed the presence of a long steady bar, which in our experiments tended to block the river right downstream branch. Similarly, in several experiments, Bertoldi and Tubino (2007) observed a steady bar with a wavelength roughly twice that of migrating bars, and which they attributed to the downstream-to-upstream morphodynamic influence of the bifurcation. In our experiments, the dynamics associated with the steady bar are far from steady. From overhead video of our experiments

(see supplemental videos), we observe dynamics at timescales on the order of minutes: the boundaries of the steady bar and scour region on the opposite wall are constantly changing, and cross-stream channels on the bar surface are constantly shifting, sometimes in the downstream direction, and sometimes in the upstream direction. The possibility of both downstream and upstream-migrating features is consistent with existing morphodynamic theory (Zolezzi and Seminara, 2001). While these dynamics are clearly not as simple as alternate bar migration, the timescales predicted by bar theories may still provide a useful reference. Additionally, as previously described, we propose that a long-period cycle associated with the steady bar occurs in conjunction with these shorter-timescale dynamics.

We were unable to define a preferred switching frequency of our experiments using a Fourier transform or wavelet analysis. A possible explanation is that our experiments do not reflect a simple superposition of frequencies, but rather represent a con-catenation of switches of variable duration. Instead, we quantified the frequency of our bifurcation experiments by compiling the intervals between threshold-crossings. A justification for this approach can be found in the asymmetry histograms of Fig. 7, showing that the bifurcation spends most of its time in an asymmetric state. This approach was inspired by work on the polarity intervals of the Earth's geomagnetic system. The standard view of polarity reversals is that they occur randomly (albeit with short-time inhibition), but that the underlying frequency of the reversal process changes through time (Merrill and McFadden, 1994). While a similar approach could be taken to modeling avulsion in our experiments, the small number of observations we have does not justify it (our data is fit by the simpler lognormal distribution). Theoretically, we do not expect downstream-controlled avulsion to be a completely random process. If the effect of the downstream-control is to create a tendency to correct asymmetries in the integrated sediment flux delivery, then the history of previous reversals should matter, even if our data set is not sufficient to define a preferred frequency.

The histograms for the net-depositional portions of experiments 4 and 5 show some indication of bimodality (Fig. 8), but as mentioned previously, due to the small total number of switches, we cannot rule out that these histograms are random realizations of an underlying lognormal distribution. In any case, we find that the range of switch durations we observe in our experiments spans the durations we would predict for migrating-bar-induced switching to downstream-controlled avulsion. Linear bar theory is known to over-predict the speed of fully-developed migrating bars (Adami et al., 2014). However, given the short length of our upstream branch, bars are unlikely to be fully developed; underdeveloped bars tend to migrate faster, and therefore more in line with linear theory (Federici and Seminara, 2003). Based on our estimates of bar migration timescales, bar migration could explain the short-duration switches we observed, but does not explain the entire distribution. Downstream-controlled avulsion could explain some of the longer-duration switches we observed. However, no existing theory explains the long-duration events we observed under bypass conditions; our theory predicts infinite duration, whereas ordinary bar migration would result in a much shorter timescale.

Previous work has tied interavulsion time to deposition rate normalized by the channel depth (Mohrig et al., 2000; Jerol-mack, 2009). Consistent with this work, we find in our experiments that deposition reduces the maximum interavulsion time, whereas bypass leads to long periods of time without avulsion. However, our work also demonstrates that net deposition is not strictly necessary for avulsion to occur. We find that upstream controls such as bar migration and lateral erosion allow for reactivation of the previously subordinate branch. Similar avulsion mechanisms have been observed in the field. Makaske et al.

(2012) presents a case study of two avulsions on the Taquari megafan, Brazil. Although both avulsions are associated with channel-belt aggradation, the authors propose that lateral migration played an important role in the mid-fan avulsion, whereas downstream controls were more important in the lower-fan avulsion. Bertoldi (2012) presents an example of an upstream-dominated bifurcation in the Tagliamento River, Italy, and found that its evolution was dominated by bar migration from upstream. In our experiments, we observe a wide range in avulsion frequencies, which we propose arises from the interplay between upstream controls (e.g. bar migration, lateral erosion) and the downstream control (deposition vs. bypass). Previous work on the Rhine-Meuse Delta, The Netherlands, shows that interavulsion time and avulsion duration are highly variable (Stouthamer and Berendsen, 2001). While Stouthamer and Berendsen (2007) proposed using interavulsion time as a way to distinguish autogenic and allogenic causes for avulsion, our work demonstrates that autogenic dynamics alone are sufficient to produce high variability in interavulsion times.

Compared to bifurcations in nature, our experimental bifurcation has a reduced degree of freedom. In nature, channels can widen, and bifurcations can change their planform geometry. Our experiments are most directly analogous to bifurcations where the bed is free to evolve, but the banks are leveed and stabilized by engineering works. Despite the relatively simple design and geometry of our experiments, we observe highly complex switching dynamics. While a freely evolving planform geometry introduces additional degrees of freedom, we predict that the downstream control should influence the qualitative dynamics in a similar way. For example, on the broader scale of whole-delta response to downstream boundary conditions, the delta experiments of Kim et al. (2013) showed that progradation results in rapid lateral migration and avulsion, whereas bypass caused the delta to consolidate its flow into a single, static channel. Similarly, Carlson et al. (2018) showed that deeper basins increase autogenic timescales in delta experiments. Although we predict similar qualitative dynamics, bifurcations that are not laterally fixed will differ quantitatively relative to our fixed-geometry experiments. In particular, if each branch distributes sediment over a nourishment area rather than just the channel itself (Shaw et al., 2016), then the progradation mass balance will change, potentially increasing the inter-avulsion timescale.

## 7  Conclusions

Based on a set of bifurcation experiments to test the influence of downstream boundary conditions (deposition vs. bypass) on the flux partitioning and its dynamics, we find:

- Bypass produces a qualitative change in avulsion dynamics compared to deposition. We observe frequent avulsions under depositional conditions, but bypass produces long periods of time where one branch dominates. The degree of asymmetry also tends to increase under bypass conditions.

- Consistent with existing theory, increasing the width-to-depth ratio of the upstream branch increases flux partitioning asymmetry.

- There are key differences between our experimental results and those of the most relevant existing model (Salter et al., 2018). With net deposition, avulsions are more irregular than those produced by the model. Under bypass conditions,

long periods where one branch dominates occur, but unlike in the model, these are not permanent. These differences appear to be related to bar effects in the upstream channel, which are not included in the model.

– The switching timescales of our experiments span a range from bar timescales on the order of minutes, to downstream-controlled avulsion timescales in the tens of minutes. However, we did not observe enough switches to define a preferred timescale or determine the underlying probability distribution.

– An additional set of experiments, run with only sand but no water, shows that in the absence of bars, bypass results in complete freezing of the dynamics in a condition where all sediment goes down one branch. These experiments, along with our fluvial experiments, suggest that downstream controls affect the dynamics in a range of physical systems, from river deltas to talus slopes.

*Code availability.* Code for the SPV model can be downloaded through CSDMS at https://csdms.colorado.edu/wiki/Model:Bifurcation.

*Data availability.* Underlying experimental data can be obtained through the Digital Repository for University of Minnesota at https://doi.org/10.13020/ngys-r052

*Video supplement.* A video supplement can be accessed through the TIB AV-Portal: https://av.tib.eu/series/639/bifurcation+experiments. The same videos are also available in the above data repository.

*Competing interests.* The authors have no competing interests to report.

*Acknowledgements.* GS acknowledges funding by the National Science Foundation Graduate Research Fellowship under Grant No. 00039202. The authors thank two anonymous reviewers for helpful comments that greatly improved the manuscript. The authors also thank Dr. Kimberly Hill and Dr. Walter Bertoldi for insightful discussion, and give special thanks to Dr. Marco Redolfi for assistance with the linear bar theory calculations.

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
