# Peer review of "How does the downstream boundary affect avulsion dynamics in a laboratory bifurcation?"

_Earth Surface Dynamics, 2019_

## Referee Comment (RC1) · Anonymous Referee #1 · 9 Jul 2019

**General comments:**

Salter et al., present results from experiments that build on their previous numerical modeling work to examine discharge partitioning through a single bifurcation. Unlike previous models by other researchers, the authors allow net deposition in the system. In the experiments, the authors observe frequent switching of water and sediment discharge between the two branches, with asymmetry increasing for increasing width-to-depth ratios and decreasing dimensionless sediment fluxes. The authors argue that discharge partitioning is strongly affected by progradation vs. bypass, as well as by bar dynamics that generate high frequency oscillations in discharge partitioning. Asymmetry is higher during bypass, as progradation is hypothesized to autocorrect strong asymmetry. In the absence of bars (sand pile experiments), the

authors find increased asymmetry and a 'frozen' stable asymmetry during bypass, as bars are not present to reroute sediment and reactivate an abandoned channel.

Overall, I find the paper very interesting and of great importance to the scientific community. The authors present a thorough explanation of their experimental methods and analysis of their results. I also find their explanations of avulsion and bar-induced switching compelling, and I appreciate their discussion of the applicability of their experiments. Below I highlight several concerns largely around the presentation of figures, results, and discussion that will strengthen the paper and clarify several key points.

**Specific major comments:**
Due to a general shortage of referencing, it is often difficult to tell which statements are the authors' own opinions and which are based on previous research/literature. For example, Page 1, Lines 23-24: Which studies? List some examples. Page 2, Lines 8-10: This is clearly based on other studies. Which ones? Include references here. The authors should also more directly relate their results to those of Bertoldi and Tubino (2007) and Bertoldi et al. (2009) (and/or others) throughout the results and discussion. This will put their results better into the broader context.

I find the language regarding the effects of 'deposition' and the 'downstream boundary' on avulsion dynamics confusing. The authors are examining how the partitioning of water and sediment down two branches of a bifurcation is affected by progradation vs. bypass. Yes, of course progradation is deposition, but 'progradation' is also more specific and informative to the reader. It also then makes it clear that the authors are not changing anything about the downstream boundary, but instead the boundary condition is changing as the system progrades. I suggest changing 'deposition' to 'progradation.' This is especially important in cases like Page 3 Line 28: 'The purpose of our experiments was to test the effect of deposition on bifurcation dynamics.' This is misleading on several fronts. First, deposition is a part of bifurcation

dynamics, so it is hard to understand how you test the effect of it. Second, it sounds like you are examining bifurcation development and evolution, when in fact you are examining discharge partitioning in a bifurcation. This needs to be explicitly stated and clarified. Finally, this distinction will really help in your discussion (e.g., Page 19, lines 17-19). You state that deposition is not necessary for avulsion, as bar migration also allows for channel reactivation. But bar migration necessarily involves 'deposition,' so this statement is not as impactful as it could be. Progradation is not necessary for avulsion, as bar migration can also induce oscillations and channel reactivation.

There is a lot of useful information in the supplemental material, but there is not one reference to it in the manuscript. The authors should reference the supplemental material where appropriate to guide readers to additional data and figures. This is especially helpful for the argument regarding estimations of Shields stresses. I understand that the experimental movies are available through the NCED data repository, but it would also be helpful to have them as supplemental information with the manuscript to make them easier to access.

Figures 4 and 5: I suggest adding some more labeling to these figures to make it easier on the reader. Authors should add text above each plot stating the channel aspect ratio and dimensionless sediment flux. I also strongly recommend calculating and plotting the mean asymmetry for each period (i.e., progradation, transition, bypass) for each experiment and adding that text to the plot. The authors describe differences in asymmetry in each period and between experiments, but quantifying it here would really drive that point home and make interpretation much easier.

Page 8, Line 16: What is the justification for saying exp 2 bypass portion does not have periods of higher asymmetry? From Fig 5, it looks like exp 1 bypass has periods of 0.55 in RL vs. 0.45 in RR. There are comparable numbers in exp 2 bypass, and occasionally an even stronger asymmetry. This needs to be clarified. Quantifying

it on the plot will also make this easier.

Page 12 first paragraph: Call readers attention to the relevant parts of Figure 6. It is hard to follow without knowing which image to look at when. Looking at 6a, it appears the deepened branch is the right one, but the authors refer to a deepening on the river left. So is the left deepening in b? This needs to be clarified.

Figure 8: Also needs labeling. Add estimated timescales from your analysis as vertical lines or asterisks or some symbol. Otherwise the reader has to find your numbers in the text and then go back to the plot and figure out where those go. Your statements regarding controls on avulsion timescales will be easier to digest if the figure is clear.

Page 15, Lines 30-33: Where do alpha and r come from? What are they? Why are they important? You introduce variables that are not defined, are not in an equation in the paper, and appear to never be used again. If this is central to your argument, then you need to elaborate. Since the timescale is not sensitive to them, why are they here? I recommend removing this and, if necessary, adding a supplemental section that elaborates on the model and any relevant sensitivity analysis.

What are the previously unrecognized long-timescale bar dynamics mentioned in the abstract? Your bar timescales described in the paper are all very short. You restate something similar on Page 18, Lines 12-14: Are you arguing bars induced the longer-period switching during bypass? Where is the explanation of this in the data? Your bar-related switching timescales were all < 3 minutes. If you are not arguing this, then you need to reword.

**Specific minor comments:**

Page 5 Lines 6-8: I don't follow this explanation of your image processing. Why were you only looking in the 0.2-0.3 m range downstream? Are you averaging the cross-stream direction within each branch? So should this be 0.02 – 0.03? This needs to be clarified in the text.

Figure 6: Also recommend labeling. Flow direction, location of bar/scour, quantified asymmetry value.

Page 16, Line 13: Why 3W? Include a reference to justify this choice. Why is this timescale so different from linear bar theory?

Page 18, Lines 3-4: So did you test all of these? More information on these experiments is necessary.

Why was the angle of 16 degrees chosen if the authors acknowledge that this is very low compared to nature?

**Technical corrections:**
Page 2 last paragraph: 'channel aspect ratio' presumably is width:depth, but please specify for the reader. This can be easily fixed by adding '(i.e., aspect ratio)' after 'width-to-depth ratio' earlier on this page.

Page 3, Line 26: Change 'We did our best to level' to 'We leveled'

Page 5, Line 16: delete comma after 'overhead images'

Page 7, Line 11: replace 'that paper' with actual reference (Salter et al., 2018)

Page 8, Line 8: 'water discharge increases **(i.e., W/h decreases)**, asymmetry decreases.'

Page 8, Line 15: Recommend adding 'where the river left branch is favored, **but the asymmetry is still small (X)**.'

Figure 3: It needs to be clarified if this is water or sediment discharge.

Page 11, Line 1-2: Does this matter? Recommend deleting.

Page 11, Lines 11-12: 'The asymmetry magnitude during these periods...' Which periods? There is a 'these periods' phrase also in the preceding sentence. Please clarify.

Page 11, Line 24: Fix parentheses 'experiments of Bertoldi et al., **(**2009).' This error also occurs elsewhere in the paper.

Figure 7: 'Histograms of the **water** discharge asymmetry' Also the asymmetry needs to be defined. It is on the horizontal axis but is never defined in the paper. RL and RR I assume are river left and river right, but this is similarly never defined anywhere.

Page 14, Line 4: This is confusing. Please reword. I assume this is meant to say that right branch-focused strong asymmetry is diminished under bypass relative to depositional phase.

Page 15, Line 25: Which model? SVP 18? Specify.

Page 16, Line 21-22: Reference needed.

Page 18, Line 10: 'deposition acts **as** a stopgap:'

Page 20, Line 15: Add Salter et al., 2018 reference for 'most relevant existing model.'

---

## Referee Comment (RC2) · Anonymous Referee #2 · 12 Jul 2019

The paper by Salter et al. reports on two sets of experiments designed to investigate bifurcation dynamics. The first set reproduces depositional and equilibrium sediment configurations, whereas the third one involves only sand (and no water) to assess the role of downstream conditions, without the upstream effect of bar migration. The experiments are well designed and monitored, the results are relevant and well explained, and the paper is written clearly with informative figures. Overall, I think that the paper is very good and I have only a few minor comments. I really like figure 7, which to my knowledge is an innovative way to present the dynamics of fluvial bifurcations and figure 8, which reports on one of the few quantifications of bifurcation timescale.

My main comment is about the length of the upstream channel and the formation of bars (Page 3, line 21-23). The length of the upstream channel is 9 times its width. This

means it is probably too short to observe free alternate bars formation and migration. There is probably space only for maximum one full wavelength. Could this have an impact on the bifurcation dynamics you observe? Did you observe migration of free alternate bars? The large bar closing one of the bifurcates is a steady bar forced by the bifurcation and I imagine it does not migrate, right? This steady bar formed also in the experiments reported in Bertoldi and Tubino, 2007. I think this issue should be discussed with more detail in the paper. Other minor comments and suggestions: Page 3, line 24. 16 degrees is a low value for the angle between the two branches. This value was probably forced by space constrains, that are often impossible to overcome. Do you have any idea whether (and how) this could impact the observed dynamics? Page 5, line 6-8. I do not understand this sentence and procedure. What is the center 0.2-0.3 m o of each branch? Page 7, line 12. The length of these cells is a crucial parameter in the model. Maybe it is worth introducing it here? Now you only find them at page 15, line 30-31, but their meaning is probably a mystery for anyone not knowing the model. Page 7, chapter 3. I miss a detail form the model. During the depositional and transitional phases, the longitudinal slope in the upstream branch is increasing. As a result the width to depth decreases and the shear stress increases, meaning that the bifurcation should increase its asymmetry (as predicted by BRT model). Does the SVP model take these changes into account? Or did you consider only the final value of the slope? Moreover, could this effect explain (at least partly) the observed differences between the transitional and the by-pass phases and the shift towards more asymmetrical configurations? Page 12, line1-13. This is an interesting observation. I wonder whether you can provide an estimate of this longer timescale. Probably it was not easy to observe many cycles, but still, it would be ggod. Are these longer cycles visible in figure 8? Is the small peak at 2ˆ8 minutes for exp. 4 - bypass related to this process? Page 15, line 22. There is no indication on how many switches you observed.

Page 15, line 22 and following paragraphs on time scale estimation. I have two observations about this. First: from your estimations it looks like roughly the time scale for exp. 5 (both considering bars and downstream control) is 2-4 time that of exp. 4,
**ESurfD**

Interactive
comment

mainly due to the lower sediment flux. It means 1-2 steps on your log scale in figure 8. It is difficult to tell (and the statistical test you performed do not allow for a quantitative assessment), but it looks reasonable, looking at the plots. Maybe you could add a comment about that. Also, would that mean that the long switching scale for exp. 5 could be 2^9-2^10 (and therefore that the experiments were not long enough?) The second point is about the timescale related to alternate bar migration. Did you really observed bar migration on the order of (a few) minutes during the experiments? As for the main comment above, with a relatively short upstream channel I do not expect formation (and migration) of free alternate bars.

---

## Author Comment (AC1) · 9 Aug 2019

**Response to reviewers on "How does the downstream boundary affect avulsion dynamics in a laboratory bifurcation?"**

Gerard Salter, Vaughan R. Voller, Chris Paola

August 9, 2019

We thank both anonymous reviewers for their helpful comments, which greatly improve the manuscript. In this document, we list the reviewers' comments in black, and our response in blue. Line numbers correspond to lines in the manuscript with highlighted changes, where P#L# denotes the page and line numbers.

**Response to Reviewer #1**

General comments: Salter et al., present results from experiments that build on their previous numerical modeling work to examine discharge partitioning through a single bifurcation. Unlike previous models by other researchers, the authors allow net deposition in the system. In the experiments, the authors observe frequent switching of water and sediment discharge between the two branches, with asymmetry increasing for increasing width-to-depth ratios and decreasing dimensionless sediment fluxes. The authors argue that discharge partitioning is strongly affected by progradation vs. bypass, as well as by bar dynamics that generate high frequency oscillations in discharge partitioning. Asymmetry is higher during bypass, as progradation is hypothesized to auto-correct strong asymmetry. In the absence of bars (sand pile experiments), the authors find increased asymmetry and a 'frozen' stable asymmetry during bypass, as bars are not present to reroute sediment and reactivate an abandoned channel.

Overall, I find the paper very interesting and of great importance to the scientific community. The authors present a thorough explanation of their experimental methods and analysis of their results. I also find their explanations of avulsion and bar-induced switching compelling, and I appreciate their discussion of the applicability of their experiments. Below I highlight several concerns largely

around the presentation of figures, results, and discussion that will strengthen the paper and clarify several key points.

Specific major comments: Due to a general shortage of referencing, it is often difficult to tell which statements are the authors' own opinions and which are based on previous research/literature. For example, Page 1, Lines 23-24: Which studies? List some examples. Page 2, Lines 8-10: This is clearly based on other studies. Which ones? Include references here. The authors should also more directly relate their results to those of Bertoldi and Tubino (2007) and Bertoldi et al. (2009) (and/or others) throughout the results and discussion. This will put their results better into the broader context.

We intended for the references on pg 2 L4-5 of the original manuscript to apply to the entire paragraph; we have now moved the references to the first sentence of the paragraph to make this more clear (P1L23). Additionally, as also requested by the second reviewer, we added an additional paragraph in the discussion section on the role of bars in our experiments. In this paragraph, we relate our results more directly to Bertoldi et al. 2007/2009 (P19L26).

I find the language regarding the effects of 'deposition' and the 'downstream boundary' on avulsion dynamics confusing. The authors are examining how the partitioning of water and sediment down two branches of a bifurcation is affected by progradation vs. bypass. Yes, of course progradation is deposition, but 'progradation' is also more specific and informative to the reader. It also then makes it clear that the authors are not changing anything about the downstream boundary, but instead the boundary condition is changing as the system progrades. I suggest changing 'deposition' to 'progradation.' This is especially important in cases like Page 3 Line 28: 'The purpose of our experiments was to test the effect of deposition on bifurcation dynamics.' This is misleading on several fronts. First, deposition is a part of bifurcation dynamics, so it is hard to understand how you test the effect of it. Second, it sounds like you are examining bifurcation development and evolution, when in fact you are examining discharge partitioning in a bifurcation. This needs to be explicitly stated and clarified. Finally, this distinction will really help in your discussion (e.g., Page 19, lines 17-19). You state that deposition is not necessary for avulsion, as bar migration also allows for channel reactivation. But bar migration necessarily involves 'deposition,' so this statement is not as impactful as it could be. Progradation is not necessary for avulsion, as bar migration can also induce oscillations and channel reactivation.

We thank the first reviewer for pointing out an area of ambiguity in our terminology. We found several instances where we used "deposition" to refer to the first phase of the experiment, which we should have called "progradation" for the sake of consistency and clarity. However, we do find it useful to retain the use of the word "deposition" to refer to the condition that the sediment supplied at the upstream boundary exceeds the sediment exiting the downstream boundary. In other words, we use "deposition" as a descriptor of the system-wide mass balance, which we contrast with "bypass", where on average the input

and output sediment fluxes are in balance (albeit with statistical fluctuations due to autogenic dynamics, e.g. bar migration). A first reason is that it is useful to distinguish the first two net depositional phases of the experiment from the bypass phase, but the word "progradation" applies only to the first phase. Secondly, our hypothesis based on the SPV numerical model is that deposition vs. bypass is the critical control, not necessarily progradation. We showed in that paper that deposition even without progradation yields ongoing avulsion dynamics. Furthermore, using the SPV model, it is possible to design a scenario where relative sea level fall allows progradation to occur without deposition upstream at the bifurcation, and under this scenario the bifurcation dynamics are frozen. Therefore, the important role of progradation in our experiments is that it results in net deposition, producing the two-way feedback between sediment flux partitioning and the downstream bed slopes. We added a sentence (P2L26) to explain that net deposition is the key control. We also modified Figure 3 by choosing a more representative simulation which illustrates that switching can continue into the transitional period (which we note is what we observed in both of the model runs used later in the paper). The reviewer rightly points out that bar migration involves deposition. However, this deposition occurs locally at the bar-scale, whereas we are referring to deposition at the system-wide scale. In order to remove this ambiguity, we have replaced "deposition" with "net deposition" or "system-wide deposition" anywhere in the text where the two meanings could potentially be confused. Additionally, we added the following text in the introduction to clarify our usage of the term, as well as clarify the importance of the downstream boundary (P4L1):

> Our usage of the term "deposition" here refers to the system-wide mass balance condition that the sediment flux input exceeds the sediment flux exiting the downstream boundary. In contrast, we define "bypass" as the condition that the sediment flux input and output are in balance on average. Because the sediment flux input from upstream is fixed, the downstream boundary controls whether the system is net-depositional or in bypass. We note that local deposition (e.g. due to bar migration) can occur even when the system is in a bypass state.

There is a lot of useful information in the supplemental material, but there is not one reference to it in the manuscript. The authors should reference the supplemental material where appropriate to guide readers to additional data and figures. This is especially helpful for the argument regarding estimations of Shields stresses. I understand that the experimental movies are available through the NCED data repository, but it would also be helpful to have them as supplemental information with the manuscript to make them easier to access.

We have added references to the supplemental info where relevant in the manuscript. We also uploaded the videos to the TIB AV-Portal, which is how ESURF handles video supplements.

Figures 4 and 5: I suggest adding some more labeling to these figures to make it easier on the reader. Authors should add text above each plot stating the channel aspect ratio and dimensionless sediment flux. I also strongly recommend calculating and plotting the mean asymmetry for each period (i.e., progradation, transition, bypass) for each experiment and adding that text to the plot. The authors describe differences in asymmetry in each period and between experiments, but quantifying it here would really drive that point home and make interpretation much easier.

Thank you for the suggestions, done.

Page 8, Line 16: What is the justification for saying exp 2 bypass portion does not have periods of higher asymmetry? From Fig 5, it looks like exp 1 bypass has periods of 0.55 in RL vs. 0.45 in RR. There are comparable numbers in exp 2 bypass, and occasionally an even stronger asymmetry. This needs to be clarified. Quantifying it on the plot will also make this easier.

We meant relative to the degree of asymmetry earlier in the same experiment. In order to clarify, we added the word "relatively" to the sentence in question (P11L17).

Page 12 first paragraph: Call readers attention to the relevant parts of Figure 6. It is hard to follow without knowing which image to look at when. Looking at 6a, it appears the deepened branch is the right one, but the authors refer to a deepening on the river left. So is the left deepening in b? This needs to be clarified.

We added references to the specific relevant subfigures througout the paragraph, which should make the explanation easier to follow. And yes, between subfigures (a) and (b) the depth of the RL branch increases.

Figure 8: Also needs labeling. Add estimated timescales from your analysis as vertical lines or asterisks or some symbol. Otherwise the reader has to find your numbers in the text and then go back to the plot and figure out where those go. Your statements regarding controls on avulsion timescales will be easier to digest if the figure is clear.

We have implemented the suggestion and agree that it greatly improves the quality of the figure.

Page 15, Lines 30-33: Where do alpha and r come from? What are they? Why are they important? You introduce variables that are not defined, are not in an equation in the paper, and appear to never be used again. If this is central to your argument, then you need to elaborate. Since the timescale is not sensitive to them, why are they here? I recommend removing this and, if necessary, adding a supplemental section that elaborates on the model and any relevant sensitivity analysis.

We added an explanation for these parameters earlier in the paper when describing the model (P7L5). Because BRT-based bifurcation models tend to be

highly sensitive to these parameters, we think it is worth mentioning briefly that we tested their effect.

What are the previously unrecognized long-timescale bar dynamics mentioned in the abstract? Your bar timescales described in the paper are all very short. You restate something similar on Page 18, Lines 12-14: Are you arguing bars induced the longer-period switching during bypass? Where is the explanation of this in the data? Your bar-related switching timescales were all < 3 minutes. If you are not arguing this, then you need to reword.

We reworded to emphasize that the long-period dynamics are associated with a steady bar; we do not attribute the long-period dynamics to the action of migrating bars (P1L12, P19L14).

Specific minor comments: Page 5 Lines 6-8: I don't follow this explanation of your image processing. Why were you only looking in the 0.2-0.3 m range downstream? Are you averaging the cross-stream direction within each branch? So should this be 0.02 - 0.03? This needs to be clarified in the text.

This was a typo, we meant 2-3 cm. We have added slightly more detail to explain this step (P5L15).

Figure 6: Also recommend labeling. Flow direction, location of bar/scour, quantified asymmetry value.

Done, and we added the quantified asymmetry values to the figure caption.

Page 16, Line 13: Why 3W? Include a reference to justify this choice. Why is this timescale so different from linear bar theory?

Reference added. The important thing is not whether it should be 3W or 5W, but rather to get a rough estimate. The simple calculation is useful because the linear bar theory calculations are much more involved. However, we found an error in the simple calculations: the corrected values for the simple bar timescale are half of the erroneous ones. We also found an unrelated error in the linear bar theory calculations. With the help of Dr. Marco Redolfi, who is an expert on linear bar theory, we revised those calculations and find that our new numbers are much more physically realistic than before. With the corrected numbers, we find that the simple bar theory timescales are short relative to the linear bar theory timescales. We therefore note (P17L12) that the assumption of no sediment bypassing the bar might explain why the simple bar calculations yield a relatively short timescale.

Page 18, Lines 3-4: So did you test all of these? More information on these experiments is necessary.

Yes. We provide some additional detail for exactly what we tested (P19L1).

Why was the angle of 16 degrees chosen if the authors acknowledge that this is very low compared to nature?

We added the following text as explanation (P3L27):

We selected a narrow angle due to space constraints and to reduce shadows in the topographic scans. We note that angle does not play a role in the quasi-1D model of Bolla Pittaluga et al. (2003), and Thomas et al. (2011) found that angle has little effect on discharge partitioning in fixed-bed experiments.

Technical corrections:

Page 2 last paragraph: 'channel aspect ratio' presumably is width:depth, but please specify for the reader. This can be easily fixed by adding '(i.e., aspect ratio)' after 'width-to-depth ratio' earlier on this page. Done.

Page 3, Line 26: Change 'We did our best to level' to 'We leveled' Done.

Page 5, Line 16: delete comma after 'overhead images' Done.

Page 7, Line 11: replace 'that paper' with actual reference (Salter et al., 2018) Done.

Page 8, Line 8: 'water discharge increases (i.e., W/h decreases), asymmetry decreases.' Done.

Page 8, Line 15: Recommend adding 'where the river left branch is favored, but the asymmetry is still small (X).' Done.

Figure 3: It needs to be clarified if this is water or sediment discharge. Done.

Page 11, Line 1-2: Does this matter? Recommend deleting.

We included it because there is discussion in other parts of the paper about how slight imperfections in the setup tended to result in RL being favored. Experiment 2 is therefore an important exception, and highlights how subtle the bifurcation sensitivity to imperfections appears to be.

Page 11, Lines 11-12: 'The asymmetry magnitude during these periods. . .' Which periods? There is a 'these periods' phrase also in the preceding sentence. Please clarify.

We added specific time intervals in the preceding sentence, and added clarification in the sentence in question (P11L27).

Page 11, Line 24: Fix parentheses 'experiments of Bertoldi et al., (2009).' This error also occurs elsewhere in the paper. Fixed.

Figure 7: 'Histograms of the water discharge asymmetry' Also the asymmetry needs to be defined. It is on the horizontal axis but is never defined in the paper. RL and RR I assume are river left and river right, but this is similarly never defined anywhere. We fix both issues at P11L4.

Page 14, Line 4: This is confusing. Please reword. I assume this is meant to say that right branch-focused strong asymmetry is diminished under bypass relative to depositional phase. Your reading of the sentence is indeed what we intended, but we improved the wording (P13L25).

Page 15, Line 25: Which model? SVP 18? Specify. Fixed.

Page 16, Line 21-22: Reference needed. Fixed.

Page 18, Line 10: 'deposition acts as a stopgap:' Fixed.

Page 20, Line 15: Add Salter et al., 2018 reference for 'most relevant existing model.' Fixed.

**Response to Reviewer # 2**

The paper by Salter et al. reports on two sets of experiments designed to investigate bifurcation dynamics. The first set reproduces depositional and equilibrium sediment configurations, whereas the third one involves only sand (and no water) to assess the role of downstream conditions, without the upstream effect of bar migration. The experiments are well designed and monitored, the results are relevant and well explained, and the paper is written clearly with informative figures. Overall, I think that the paper is very good and I have only a few minor comments. I really like figure 7, which to my knowledge is an innovative way to present the dynamics of fluvial bifurcations and figure 8, which reports on one of the few quantifications of bifurcation timescale.

My main comment is about the length of the upstream channel and the formation of bars (Page 3, line 21-23). The length of the upstream channel is 9 times its width. This means it is probably too short to observe free alternate bars formation and migration. There is probably space only for maximum one full wavelength. Could this have an impact on the bifurcation dynamics you observe? Did you observe migration of free alternate bars?

Yes, we observed free bar migration. We added an example of overhead images showing bar migration on timescales on the order of minutes to the supplemental info. Most likely these bars are not fully developed, but they appear to influence the flow partitioning nevertheless. During other parts of the experiment (such as the majority of the bypass portion of experiment 4), the presence of migrating bars is not as obvious, but nevertheless downstream-migrating features can be observed from the overhead videos. Even if it is wrong to consider these features as related to migrating bars, linear bar theory may provide a useful reference timescale. After revising our calculations based on linear bar theory, we find that the predicted time scales line up well with the short time scale peaks in our histograms. If the upstream branch were longer, we might expect the migrating bars to develop more fully. Fully developed bars migrate more slowly, so the timescale would increase. Additionally, an increase in bar height could enhance the influence of bars.

The large bar closing one of the bifurcates is a steady bar forced by the bifurcation and I imagine it does not migrate, right? This steady bar formed also

in the experiments reported in Bertoldi and Tubino, 2007. I think this issue should be discussed with more detail in the paper.

Yes, this is a steady bar, and it was always located upstream of the river right branch. It did not migrate except to the extent that the boundaries of the bar tend to fluctuate. We added a paragraph to the discussion section on the dynamics we observed associated with the steady bar and migrating bars, in the context of the previous work by Bertoldi et al. (2007/2009) (P19L26).

Other minor comments and suggestions: Page 3, line 24. 16 degrees is a low value for the angle between the two branches. This value was probably forced by space constrains, that are often impossible to overcome. Do you have any idea whether (and how) this could impact the observed dynamics?

The first reviewer makes the same point, which we now address. We cite Thomas et al. (2011), who found that angle did not significantly affect flow partioning. In our view, the effect of angle is most likely second order; the dynamics will most likely remain qualitatively similar regardless of angle. We added a sentence of explanation (P3L27).

Page 5, line 6-8. I do not understand this sentence and procedure. What is the center 0.2-0.3 m o of each branch?

This was a typo, we meant 2-3 cm. We have added slightly more detail to explain this step (P5L15).

Page 7, line 12. The length of these cells is a crucial parameter in the model. Maybe it is worth introducing it here? Now you only find them at page 15, line 30-31, but their meaning is probably a mystery for anyone not knowing the model.

We added a sentence in the model explanation on the role of $\alpha$ and $r$ in the model, which addresses this point as well as the point made by the first reviewer (P7L5).

Page 7, chapter 3. I miss a detail form the model. During the depositional and transitional phases, the longitudinal slope in the upstream branch is increasing. As a result the width to depth decreases and the shear stress increases, meaning that the bifurcation should increase its asymmetry (as predicted by BRT model). Does the SVP model take these changes into account? Or did you consider only the final value of the slope? Moreover, could this effect explain (at least partly) the observed differences between the transitional and the by-pass phases and the shift towards more asymmetrical configurations?

Yes, the change in the upstream slope is captured automatically in the model. We find that the slope typically increases by a little over 10% between the start of the progradation phase and the bypass phase. As the reviewer points out, this means the aspect ratio increases, which tends to increase the asymmetry. Because depth is inversely proportional to slope to the 1/3 power, the change in depth is less than 5%. The Shields stress is proportional to the slope to the

2/3 power. An increase in the Shields stress tends to decrease the asymmetry, meaning that changing the slope introduces two partially offsetting effects on the asymmetry. We used the BRT model (i.e. not including deposition) to check how much the change in the upstream parameters due to the slope change affects the asymmetry, independent of deposition. Whether the aspect ratio or Shields stress wins depends on the value of $\alpha$, which makes sense when considering the neutral stability curve reported by Bolla Pittaluga et al. (2015). In general, we find that the effect on the asymmetry is small. In our model runs, the change in asymmetry is therefore dominated by the strength of the downstream deposition feedback. We added clarification that the upstream branch is considered explicitly as part of the model runs (P7L9).

As to whether this affected our experiments, we find an 11% slope increase in experiment 2, and no appreciable changes in the usptream slope in any of the other experiments. In all our experiments, we do not detect a change in the upstream channel depth. Therefore, we rule out changes in the upstream slope as the source of the asymmetry changes we observe, and conclude that the downstream control is the best explanation.

Page 12, line1-13. This is an interesting observation. I wonder whether you can provide an estimate of this longer timescale. Probably it was not easy to observe many cycles, but still, it would be ggod. Are these longer cycles visible in figure 8? Is the small peak at $2^8$ minutes for exp. 4 - bypass related to this process? Page 15, line 22. There is no indication on how many switches you observed.

We observe around 2.5 cycles in experiment 4. In Figure 5, the first cycle starts around minute 1000, the second cycle starts around minute 1600, and the start of the third cycle is around minute 2200. The small peak in figure 8 at $2^8$ minutes is only 3 switches, but yes, they are associated with the cycles (although they don't take up the full duration of a cycle). P13L14 we add a sentence stating how many cycles we observed.

Page 15, line 22 and following paragraphs on time scale estimation. I have two ob- servations about this. First: from your estimations it looks like roughly the time scale for exp. 5 (both considering bars and downstream control) is 2-4 time that of exp. 4, mainly due to the lower sediment flux. It means 1-2 steps on your log scale in figure 8. It is difficult to tell (and the statistical test you performed do not allow for a quantitative assessment), but it looks reasonable, looking at the plots. Maybe you could add a comment about that.

Thank you for the interesting observation; we added a comment on this (P17L16).

Also, would that mean that the long switching scale for exp. 5 could be $2^9 - 2^{10}$ (and therefore that the experiments were not long enough?)

We presented an argument in the paper/supplemental info for why we believe the complete avulsion was most likely permanent (the scour-widening mechanism that reopened branch RR during experiment 4 occured but did not result in a

reopening of branch RR, because the bed level in RR was significantly higher than the water level), but the reviewer is correct that running the experiment longer would make this point more convincing. We note that if a switch were to occur at the end of the experiment 5 run time, this would yield a switching time of $2^9$ minutes.

The second point is about the timescale related to alternate bar migration. Did you really observed bar migration on the order of (a few) minutes during the experiments? As for the main comment above, with a relatively short upstream channel I do not expect formation (and migration) of free alternate bars.

Yes, as mentioned above, we observed bar migration on the timescale of minutes. We agree that it is surprising given the short length of the upstream branch; in fact we originally chose the upstream branch length to be long enough to accomodate reasonable values of $\alpha$ but short enough to avoid the formation of bars. However, we found that it is impossible to understand the dynamics we observed without accounting for the influence of bars (or at least proto-bars).